Temporal changes in nasopharyngeal carriage of Streptococcus pneumoniae serotype 1 genotypes in healthy Gambians before and after the 7-valent pneumococcal conjugate vaccine

Ebruke Chinelo 1 2
Roca Anna 1
Egere Uzochukwu 1
Darboe Ousainou 1
Hill Philip C. 3
Greenwood Brian 2
Wren Brendan W. 2
Adegbola Richard A. 1 4
Antonio Martin 1 2 5 mantonio@mrc.gm
1 Vaccinology Theme, Medical Research Council Unit , Banjul , The Gambia
2 Faculty of Infectious and Tropical Diseases, London School of Hygiene & Tropical Medicine , London , United Kingdom
3 Centre for International Health, School of Medicine, University of Otago , New Zealand
4 GlaxoSmithKline Biologicals Wavre , Belgium
5 Microbiology and Infection Unit, Warwick Medical School, University of Warwick , Coventry , United Kingdom
Parish Tanya
Electronic publication date: 2015 Apr 30
Publication date: 2015
Volume: 3
Electronic Location ID: e903
Received 2014 Nov 12; Accepted 2015 Mar 31
Copyright: © 2015 Ebruke et al.
Copyright year: 2015
Copyright holder: Ebruke et al.
License: This is an open access article distributed under the terms of the Creative Commons Attribution License, which permits unrestricted use, distribution, reproduction and adaptation in any medium and for any purpose provided that it is properly attributed. For attribution, the original author(s), title, publication source (PeerJ) and either DOI or URL of the article must be cited.
License URL: https://creativecommons.org/licenses/by/4.0/

Keywords: Invasive pneumococcal disease, Nasopharyngeal Swab, Streptococcus pneumoniae serotype 1, Nasopharyngeal carriage, 7-valent pneumococcal conjugate vaccine, Gambia, Multilocus Sequence Typing, Sequence type, ST217 hyper virulent clonal complex, Molecular epidemiology

Funding: Support for Chinelo Ebruke’s PhD studentship and research costs was provided by the Medical Research Council Unit, The Gambia. The funders had no role in study design, data collection and analysis, decision to publish, or preparation of the manuscript.

==============================
Streptococcus pneumoniae serotype 1 is one of the leading causes of invasive pneumococcal disease. However, this invasive serotype is hardly found in nasopharyngeal asymptomatic carriage and therefore large epidemiological studies are needed to assess the dynamics of serotype 1 infection. Within the context of a large cluster randomized trial conducted in rural Gambia to assess the impact of PCV-7 vaccination on nasopharyngeal carriage, we present an ancillary analysis describing the prevalence of nasopharyngeal carriage of pneumococcal serotype 1 and temporal changes of its more frequent genotypes. Nasopharyngeal swabs (NPS) were collected before PCV-7 vaccination (December 2003–May 2004) and up to 30 months after PCV-7 vaccination. The post-vaccination time was divided in three periods to ensure an equal distribution of the number of samples: (1) July 2006–March 2007, (2) April 2007–March 2008 and (3) April 2008–Feb 2009. S. pneumoniae serotype 1 were genotyped by MLST. Serotype 1 was recovered from 87 (0.71%) of 12,319 NPS samples collected. In the pre-vaccination period, prevalence of serotype 1 was 0.47% in both study arms. In the post-vaccination periods, prevalence in the fully vaccinated villages ranged between 0.08% in period 1 and 0.165% in period 2, while prevalence in partly vaccinated villages was between 0.17% in period 3 and 1.34% in period 2. Overall, four different genotypes were obtained, with ST3081 the most prevalent (60.71%), followed by ST618 (29.76%). ST3081 was found only in post-vaccination period 2 and 3, while ST618 had disappeared in post-vaccination period 3. Distribution of these major genotypes was similar in both study arms. Emergence of ST3081 and concomitant disappearance of ST618 may suggest a change in the molecular epidemiology of pneumococcal serotype 1 in this region. This change is not likely to be associated with the introduction of PCV-7 which lacks serotype 1, as it was observed simultaneously in both study arms. Future population-based epidemiological studies will provide further evidence of substantive changes in the pneumococcal serotype 1 epidemiology and the likely mechanisms.

Introduction

Streptococcus pneumoniae is usually found in the nasopharynx of healthy individuals, which is considered a necessary step preceding invasive pneumococcal disease (IPD), including pneumonia, meningitis, and bacteraemia (Gleich et al., 2000; Bogaert, De Groot & Hermans, 2004; Baker et al., 2005; Adegbola et al., 2006; Hill et al., 2006; Ahern & Raszka, 2009; Balicer et al., 2010). There are over 90 different S. pneumoniae serotypes of which serotype 1 is a common cause of IPD worldwide, with particular high rates of disease in sub-Saharan Africa (Hausdorff et al., 2000; Adegbola et al., 2006; Gessner, Mueller & Yaro, 2010).

As cases of invasive disease represent only a small fraction of the pneumococcal burden, there is an increasing interest on evaluating the prevalence of pneumococcal asymptomatic carriage in the nasopharynx, since this is important in understanding the dynamics of disease and transmission as well as providing a basis for assessing the impact of interventions (Bogaert, De Groot & Hermans, 2004). Interestingly, serotype 1 is rarely found in the nasopharynx of healthy individuals with prevalence always below 1%–2% (Hill et al., 2006; Laval et al., 2006; Antonio et al., 2008; Nunes et al., 2008). As a result, there are just few studies evaluating the dynamics of serotype 1 in nasopharyngeal carriage as large epidemiological studies are needed.

Although PCV-7, the first licensed pneumococcal vaccine, did not include serotype 1 in its formulation, new PCV licensed vaccines (PCV-10 and PCV-13) include this serotype and therefore understanding the dynamics of serotype 1 carriage has become a priority. Within the context of a cluster-randomized trial conducted in rural Gambia (Roca et al., 2011), we collected a large number of NPS (12, 319 samples) before and up to 30 months after starting the trial (Hill et al., 2006; Roca et al., 2011). As an ancillary study of the trial, we describe the dynamics of pneumococcal serotype 1 nasopharyngeal carriage within a period of 6 years before and after the PCV-7 trial.

Materials and Methods

Study design and collection of isolates

This study was an ancillary study of large pneumococcal carriage studies conducted in 21 selected villages in rural Gambia as previously described (Hill et al., 2006; Roca et al., 2011). Firstly, a pre-vaccination cross sectional survey was conducted between December 2003 and May 2004 in which NP samples were collected from subjects of all age groups (Hill et al., 2006). Following this, a single-blind, cluster-randomized (by village) trial to evaluate the impact of PCV-7 on pneumococcal carriage was conducted in the study villages (Roca et al., 2011). In one group of 11 villages, all individuals over the age of 30 months received one dose of PCV-7, whilst subjects in this age group resident in 10 control villages received one dose of serogroup C meningococcal conjugate vaccine. All children less than 30 months of age in both study groups and infants born during the course of the trial received PCV-7. The trial showed a marked decrease of nasopharyngeal carriage of vaccine type (VT) pneumococci in all age groups and both study arms, with a more marked drop in villages where the whole community had received PCV-7 (Roca et al., 2011). There was little change in the overall prevalence of NVT carriage following introduction of the vaccine (Roca et al., 2011).

NP swabs were collected as part of several ongoing studies. First, a large pre-vaccination cross-sectional survey (CSS) conducted in 2003–2004, prior to a PCV-7 randomized trial (Hill et al., 2006). Later, as part of the cluster-randomized PCV-7 trial, NPS were collected in three different cross-sectional surveys (from 4–6 months, 12–14 months and up to 30 months) (Roca et al., 2011) and as well as a longitudinal study between 4 and 30 months (unpublished data) after vaccination (Table 1). For the purposes of this analysis, post-vaccination data were shown stratified in three different time-periods defined to ensure at least 2,000 samples in each period as the number of samples were not equally distributed throughout the follow up period (Table 1).

Table 1 Prevalence of nasopharyngeal pneumococcal serotype 1 carriage between pre-vaccination study period and each of the post-vaccination study periods in The Gambia.

Study period	Number of NPS	Number of serotype 1 isolates (%)	P value	
Pre-vaccination period (December 2003–May 2004)	2,746	13 (0.47)	<0.001	
Post vaccination Period 1 (July 2006–March 2007)	3,986	9 (0.23)		
Post vaccination Period 2 (April 2007–March 2008)	3,469	52 (1.50)		
Post vaccination Period 3 (April 2008–Feb 2009)	2,118	13 (0.61)		
Total	12,319	87 (0.71)		

Approval for this study was obtained from the Joint Medical Research Council (MRC)/Gambia Government Ethics Committee and the Ethics Committee of the London School of Hygiene & Tropical Medicine, UK (SCC number 1032, ISRCTN 51695599). Community and individual consent was obtained from study participants and the conduct of the trial was guided by a Data Safety and Monitoring Board.

Serotyping and multilocus sequence typing (MLST)

A total of 87 S. pneumoniae serotype 1 isolates obtained from NP swabs during a survey were identified by latex agglutination (Hill et al., 2006) and confirmed by molecular serotyping (Morais et al., 2007). Multilocus sequence typing was performed on viable S. pneumoniae serotype 1 isolates recovered after storage at −70 °C as previously described (Antonio et al., 2008).

Data analysis

All statistical analysis were carried out in STATA (version 11; StataCorp, College Station, Texas, USA) using Chi-square tests. p-Values less than 0.05 were taken to indicate statistical significance. Wet season was considered from June to October each year. Sequences were edited and aligned using the Laser Gene DNA star 7.1 software. Sequence type (ST) was obtained by submission of sequences onto the MLST database website.

Results

A total of 12,319 NP samples were collected during the study: 22.2% of which were from the pre-vaccination period and 32.4%, 28.2% and 17.2% from the post-vaccination study periods 1–3, respectively. The median age of sampled individuals was 15 years (IQR 5.9—45 years), 11 years (IQR 4.7—29 years), 11 years (IQR 5.6—27 years) and 14 years (IQR 6.7—33 years), in the pre-vaccination and post vaccination periods 1–3 respectively. The overall prevalence of S. pneumoniae in the pre-vaccination period was 71.78% (1,971 out of 2,746 samples). The overall prevalence of S. pneumoniae in the post-vaccination periods 1–3 was 47.08% (4,507 out of 9,573 samples).

The overall prevalence of S. pneumoniae serotype 1 was 0.71% (87 of 12,319 samples collected). Prevalence of serotype 1 carriage was highest (1.02%) among children aged 5–14 years (p < 0.001) compared to other age groups. Serotype 1 carriage prevalence was highest in post-vaccination period 2 (1.50%) compared to the other study periods (p < 0.001) (Table 1).

Serotype 1 isolates were likely to be found during the wet season 0.81% (40 out of 4,909 samples collected) compared to the dry season 0.64% (47 out of 7,323 samples collected) (p = 0.268) but these were not statistically significant.

The overall prevalence of serotype 1 pneumococcal carriage was similar in vaccinated and control villages (0.73% vs. 0.68%; p = 0.703). In the pre-vaccination study period, prevalence of carriage of serotype 1 was the same in both vaccinated and control villages (0.47% each). However, the prevalence was lower in vaccinated than in control villages in post-vaccination period 1 (0.08% vs. 0.48%, p = 0.011), similar in vaccinated and control villages in period 2 (1.65% vs. 1.34%, p = 0.459) and higher in vaccinated villages in period 3 (1.16% vs. 0.17%, p = 0.004) (Table 2).

Table 2 Prevalence of nasopharyngeal pneumococcal serotype 1 carriage between control and vaccinated villages in each study period in The Gambia.

Study period	Village group	Number of NPS	Number of serotype 1 isolates (%)	P value	
Pre-vaccination period	Control	1,271	6 (0.47)	0.992	
	Vaccinated	1,475	7 (0.47)		
Post vaccination period 1	Control	1,468	7 (0.48)	0.011	
	Vaccinated	2,518	2 (0.08)		
Post vaccination period 2	Control	1,711	23 (1.34)	0.459	
	Vaccinated	1,758	29 (1.65)		
Post vaccination period 3	Control	1,171	2 (0.17)	0.004	
	Vaccinated	947	11 (1.16)		

MLST analysis was performed for 84 of the 87 serotype 1 isolates obtained (97%). Four different STs were obtained, with ST3081 being the predominant ST (60.71%) in both vaccinated and control villages followed by ST618 (29.76%), ST217 (7.14%) and ST303 (2.38%). Prevalence of different STs was not associated with age groups (p = 0.368). However, the distribution of STs differed over the course of the study (p < 0.001). ST3081 was seen only in the post vaccination periods. ST618 was seen in the pre-vaccination and periods 1 and 2 post vaccination, but not in the post vaccination period 3 (Fig. 1). Differences in the distribution of ST over the study periods was apparent in both vaccinated (p = 0.002) and control (p = 0.021) villages (Fig. 2), with the observed expansion of ST3081 and the disappearance of ST618 occurring in both groups (Figs. 1 and 2).

Figure 1 Distribution of Streptococcus pneumoniae 1 genotypes across study periods in The Gambia.

Figure 2 Distribution of Streptococcus pneumoniae 1 genotypes across study periods in (a) control and (b) vaccinated villages in The Gambia.

Discussion

Given that pneumococcal serotype 1 is one of the common cause of IPD worldwide and the paradox of its’ rarity in nasopharyngeal carriage, it is not unsurprising that only a few published studies have evaluated serotype 1 carriage patterns. To our knowledge, this is the largest study evaluating the dynamics of pneumococcal serotype 1 carriage. We present findings from as many as 87 serotype 1 isolates and report on the prevalence and dominant genotype patterns over a 6 year period. The finding of 0.71% overall prevalence in carriage of serotype 1 agrees with earlier findings indicating the rarity of serotype 1 in carriage studies (Brueggemann & Spratt, 2003; Hausdorff, Feikin & Klugman, 2005; Laval et al., 2006; Nunes et al., 2008; Smith-Vaughan et al., 2009). We also note that this low carriage rate was observed in both the pre- and post vaccination periods with no significant differences between study arms. However, prevalence of serotype 1 carriage was highest in the age group 5–14 years. Findings from other studies suggest that this age group is at particular risk for serotype 1 IPD as opposed to other serotypes (Adegbola et al., 2006; Gessner, Mueller & Yaro, 2010).

Introduction of the pneumococcal conjugate vaccine PCV-7 is associated with a reduction in carriage of VT serotypes but has also been linked to an increase in carriage of NVT in some settings (Mbelle et al., 1999; Huang et al., 2005; O’Brien et al., 2007) but not in our setting (Roca et al., 2011) and elsewhere (Millar et al., 2008; Roca et al., 2011). In this study, serotype 1 prevalence showed variation over the study period, but this is not likely to have been related to vaccine introduction as there was no consistent trend and no consistent difference between vaccinated and control villages. A higher carriage rate in the vaccinated group compared to the controls was observed in only one study period, and a reverse picture was observed in another post vaccination period of the study. This pattern appears more likely to be due to natural variation over time rather than to an increase in NVT serotypes due to community vaccination with PCV-7.

All STs obtained in this study belong to the ST217 hyper virulent clonal complex responsible for several epidemic outbreaks in West Africa (Leimkugel et al., 2005; Yaro et al., 2006; Antonio et al., 2008). The prevalence of the predominant serotype 1 genotypes (ST3081 and ST618) varied significantly over the study period. In period 3, we were unable to detect ST618, but noted instead the predominance of its quadruple locus variant ST3081. It is plausible that the changes between ST618 and ST3081 in this study population provide initial evidence of an expansion of the ST217 clonal complex. However, this finding could possibly have been due to temporal changes. On-going invasive pneumococcal disease surveillance studies in The Gambia such as IPD surveillance in the Upper River Region (Mackenzie et al., 2012) as well as the pneumococcal disease surveillance in the West Africa region consortium (M Antonio, pers. comm., 2015) will provide more answers to these questions. We have also shown, in our study area, the detection of a new sequence type in The Gambia without evidence that this was associated with vaccination with PCV-7. Such emergence of ST suggests natural variation in the molecular epidemiology of the pneumococcus that requires further evaluation. A report from Brazil of a study that looked at invasive serotype 1 isolates over 3 decades found temporal changes in pulse field gel electrophoresis subtypes and STs over time, but the effect of pneumococcal vaccination was not evaluated (Chiou et al., 2008). This should be closely monitored in The Gambia in the near future, as the wider PCV formulation (PCV-13) has recently been introduced as part of the Expanded Programme of Immunization.

However, we acknowledge some limitations with this study. Firstly, the samples in this study were of modest size and a larger sample size would have allowed for more robust analysis between the comparison groups. The modest number we got after sampling such a large population goes to support the notion that serotype 1 is rare in carriage. Obtaining a much larger sample size will therefore require very large epidemiological studies and its attendant challenges. Secondly, this study was limited to carriage isolates from the Western division of The Gambia. It is unclear if observations from this group are applicable to a more heterogeneous population. There is therefore a need for further studies in The Gambia, including population-based molecular epidemiological studies assessing the distribution of these STs causing IPD and whole-genome comparisons in order to identify genetic differences that could correspond with the observed differences between otherwise highly similar strains, and such studies are currently underway (Williams et al., 2012).

Conclusions

In conclusion, in this study we show the prevalence of pneumococcal serotype 1 carriage as well as the predominant genotypes and how they varied over the study periods, but this did not seem related to community vaccination with PCV-7. This provides important baseline data for further evaluation of nasopharyngeal carriage after PCV-13 has been introduced in The Gambia.

We would like to thank all the individuals who participated in the study. We also acknowledge the use of the core sequencing facility at MRC Unit, The Gambia and the S. pneumoniae MLST database which is housed at Imperial College, London, UK.

Abbreviations

IPD Invasive Pneumococcal Disease

PCV-7 7 valent pneumococcal conjugate vaccine

NPS Nasopharyngeal Swab

MLST Multilocus Sequence Typing

ST Sequence type

Additional Information and Declarations

Competing Interests

Author Contributions

Human Ethics

DNA Deposition

Richard A. Adegbola is an employee of GlaxoSmithKline Vaccines.

Chinelo Ebruke conceived and designed the experiments, performed the experiments, analyzed the data, wrote the paper, prepared figures and/or tables, reviewed drafts of the paper.

Anna Roca conceived and designed the experiments, analyzed the data, prepared figures and/or tables, reviewed drafts of the paper.

Uzochukwu Egere and Ousainou Darboe contributed reagents/materials/analysis tools, reviewed drafts of the paper.

Philip C. Hill, Brian Greenwood, Brendan W. Wren and Richard A. Adegbola conceived and designed the experiments, reviewed drafts of the paper.

Martin Antonio conceived and designed the experiments, analyzed the data, wrote the paper, prepared figures and/or tables, reviewed drafts of the paper.

The following information was supplied relating to ethical approvals (i.e., approving body and any reference numbers):

Approval for this study was obtained from the Joint Medical Research Council (MRC)/Gambia Government Ethics Committee, and the Ethics Committee of the London School of Hygiene & Tropical Medicine, UK (SCC number 1032, ISRCTN 51695599). The conduct of the trial was guided by a Data Safety and Monitoring Board, and community and individual consent was obtained from study participants.

The following information was supplied regarding the deposition of DNA sequences:

Multilocus sequencing typing (MLST) database: http://spneumoniae.mlst.net/.

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
