# Peer review of "Temporal changes in nasopharyngeal carriage of Streptococcus pneumoniae serotype 1 genotypes in healthy Gambians before and after the 7-valent pneumococcal conjugate vaccine"

_PeerJ, doi:10.7717/peerj.903_

## Round 0.1 · original submission · Major Revisions

· Academic Editor

Major Revisions

Please would you provide a revision that addresses the comments from the reviewers. In particular the comments from reviewer 2 regarding the source of the isolates.

·

Basic reporting

The investigators provide a report of a cluster randomized study of the nasopharyngeal carriage of S.pneumoniae serotype 1 within the context of an efficacy trial of a pneumococcal conjugate vaccine. The framework of a large population surveillance with carriage data from over 12,000 samples with pre vaccination and post vaccination sampling as long as 30months provides a robust dataset for analysis.

The possibility that the emergence of ST3081 and concomitant disappearance of ST618 may suggest a change in the molecular epidemiology of pneumococcal serotype 1 in this region cannot be discounted but there is still the possibility that the observed trend is temporal and not vaccine-related.

1. a). Are the observation periods defined by some objective criteria or arbitrary?
b). Were any attempts made to review the data based on seasonality (e.g. wet vs. dry season)?

2. In the study method as described by the investigators, in one group of 11 villages, all individuals over the age of 30 months received one dose of PCV-7 whilst subjects in this age group resident in 10 control villages received one dose of serogroup C meningococcal conjugate vaccine-

a). What impact on carriage can be expected of 1-dose of the conjugate vaccine?

Experimental design

The overall prevalence of S. pneumoniae serotype 1 was 0.71% (87 of 12,319 samples collected). Prevalence of serotype 1 carriage was highest (1.02%) among children aged 5 to 14 years (p<0.001) compared to other age groups and in post-vaccination period 2 (1.50%) compared to the other study periods (p<0.001) (Table 1).
a). Does this age group and the timing of peak prevalence of carriage serotype 1 correlate in any way with the incidence of invasive pneumococcal disease caused by serotype 1 from the population studied?

b). Are there any serology data from this age group during this period to improve understanding of this phenomenon?

Validity of the findings

No Comments

Comments for the author

Perhaps future studies should focus on the entire microbial community of the nasopharynx and not just a single pneumococcal serotype, since the changes following vaccination and the consequent immune pressure may primarily impact pneumococci but with a secondary effect on other pathogens and commensals

Reviewer 2 ·

Basic reporting

Overall, the paper provides some interesting results primarily on genotype changes within serotype 1 over time from carriage, in the presence of vaccine use. The authors describe using isolates from several previously well described carriage studies published by Hill et al 2006 and Roca et al 2011 where serotypes and AST were described in detail. After carefully reading the paper and looking at previous studies published from the Gambia by this group, I cannot figure out from Methods and data presented where all these isolates come from? The 13 pre-PCV isolates are described in Hill et al 2006 and not sure if these are the same described in pre-PCV arm from Roca et al 2011 paper (although Table 5 only mentions 8 serotype 1 isolates)? I can’t see where the post-PCV serotype 1 isolates come from? Roca et al 2011 paper only has 13 in 3 post-PCV study periods? So where does the additional 61 isolates come from? Then the authors mention on page 7, lines 103-106 that isolates come from cross-sectional surveys described by Roca et al 2011 and unpublished data from a longitudinal study. I assume this is a different study to the Hill et al 2010 paper since no serotype 1 isolates were reported in that study? What study is this and if isolates are used from this study do they match the study periods from the cross-sectional study? If isolates are used from the longitudinal study are there multiple NP swabs from individual persons and is more than one serotype 1 isolate per person included? It makes the paper difficult to review and comment on statistical significance if the reader can’t figure out where the isolates are coming from.

Experimental design

Study design and research questions seem sound but would need clarity on the isolate selection for the study.

Validity of the findings

Again, would need some additional information on studies isolates are coming from to make any comments on validity and statistics of the results.

Comments for the author

Abstract, page 2, line 24. Please correct the percentage for ST618.

Materials and Methods, page 6, line 88. Put a space between “in21”

Materials and Methods, page 7, line 125. Authors refer to using eBURST, I don’t see any eBURST analyses in the results, only mention of STs?

Results, page 8, lines 137-140. Table 1 doesn’t show the data that you present here? I don’t see breakdown by age?

Table 1 and 2. Heading lists isolates coming from cross-sectional studies? But Materials and Methods included longitudinal study (unpublished data)?

Figure 2. Use “Control Villages” instead of “Controlled Villages”

Discussion, page 10, lines 194-195. What kinds of additional studies would provide more clarity on the expansion of the ST217 clone? The ST3081 is a SLV of ST217 a well-documented serotype 1 clone in Africa. More carriage studies or perhaps some comparisons with serotype 1 IPD isolates from Gambia?

Discussion, page 11, lines 205-216. The authors suggest a limitation is the small number of serotype 1 isolates from carriage and that larger carriage studies are needed to look at serotype 1. Since serotype 1 is so common in Africa in IPD wouldn’t an analysis of genotypes of serotypes 1 from IPD over the same study period be useful. Are IPD serotype 1 isolates available to look at?

Discussion, page 11, line 213-214. The authors suggest using WGS to look at genetic differences – a paper was published by authors on ST618 isolate which should be referenced.

---

## Round 0.2 · accepted · Accept

· Academic Editor

Accept

Revisions accepted and paper is recommended for publication.

·

Basic reporting

The investigaotors have in this re-submission addressed as best as possible, the issues raised from the initial submission.

Experimental design

No comments

Validity of the findings

No comments

Comments for the author

No further comments. Previous concerns with methodology and sampling have been addressed in this resubmission.

Reviewer 2 ·

Basic reporting

The authors has adequately addressed all reviewers comments from the first draft.

Experimental design

The authors has adequately addressed all reviewers comments from the first draft.

Validity of the findings

The authors has adequately addressed all reviewers comments from the first draft.

Comments for the author

The authors has adequately addressed all reviewers comments from the first draft.